# Detection of Extremely Low Level Ciguatoxins through Monitoring of Lithium Adduct Ions by Liquid Chromatography-Triple Quadrupole Tandem Mass Spectrometry

**DOI:** 10.3390/toxins16040170

**Published:** 2024-03-27

**Authors:** Manami Kobayashi, Junichi Masuda, Naomasa Oshiro

**Affiliations:** 1Shimadzu Corporation, 3-25-40, Tonomachi, Kawasaki-Ku, Kawasaki 210-0821, Kanagawa, Japan; junichi@shimadzu.co.jp; 2Division of Biomedical Food Research, National Institute of Health Sciences, 3-25-26 Tonomachi, Kawasaki-Ku, Kawasaki 210-9501, Kanagawa, Japan; n-oshiro@nihs.go.jp

**Keywords:** ciguatera poisoning, ciguatoxin, liquid chromatography-triple quadrupole tandem mass spectrometer (LC-MS/MS), multiple reaction monitoring (MRM)

## Abstract

Ciguatera poisoning (CP) is the most common type of marine biotoxin food poisoning worldwide, and it is caused by ciguatoxins (CTXs), thermostable polyether toxins produced by dinoflagellate *Gambierdiscus* and *Fukuyoa* spp. It is typically caused by the consumption of large fish high on the food chain that have accumulated CTXs in their flesh. CTXs in trace amounts are found in natural samples, and they mainly induce neurotoxic effects in consumers at concentrations as low as 0.2 µg/kg. The U.S. Food and Drug Administration has established CTX maximum permitted levels of 0.01 µg/kg for CTX1B and 0.1 µg/kg for C-CTX1 based on toxicological data. More than 20 variants of the CTX1B and CTX3C series have been identified, and the simultaneous detection of trace amounts of CTX analogs has recently been required. Previously published works using LC-MS/MS achieved the safety levels by monitoring the sodium adduct ions of CTXs ([M+Na]^+^ > [M+Na]^+^). In this study, we optimized a highly sensitive method for the detection of CTXs using the sodium or lithium adducts, [M+Na]^+^ or [M+Li]^+^, by adding alkali metals such as Na^+^ or Li^+^ to the mobile phase. This work demonstrates that CTXs can be successfully detected at the low concentrations recommended by the FDA with good chromatographic separation using LC-MS/MS. It also reports on the method’s new analytical conditions and accuracy using [M+Li]^+^.

## 1. Introduction

Ciguatera poisoning (CP) is a type of food poisoning known to cause human illness due to the consumption of seafood contaminated with ciguatoxins (CTXs) [1,2]. CTXs bind to receptor site-5 in the α-subunit of the voltage-gated sodium channel, resulting in hyperexcitability of the nerve membrane [3]. Therefore, the symptoms of CP are diverse and include gastrointestinal, neurological, and cardiovascular symptoms. It is one of the world’s largest types of food poisoning involving natural toxins, and is estimated to result in between 50,000 and 500,000 cases annually, which has been proposed by the FAO and WHO 2020 [2]. CP occurs mainly in tropical and subtropical regions of the Pacific and Indian Oceans as well as the Caribbean Sea [4]. Based on the skeletal structures of CTX analogs, they are classified into four groups including CTX4A, CTX3C, C-CTX (Caribbean CTX), and I-CTX (Indian Ocean CTX) derivatives [4,5,6], in which the structure of I-CTX has not yet been elucidated. Recently, there have been reports of CP occurring in Macaronesia (the Canary Islands, Spain, and Madeira Islands, Portugal) in the East Atlantic Ocean, known as a non-CP endemic region [7,8], so there is concern that the areas affected by CP may expand [9].

In the Pacific Ocean, it has been confirmed that CTXs are produced by epiphytic dinoflagellates of the genera *Gambierdiscus* and *Fukuyoa*, which inhabit the surfaces of macroalgae [4]. These dinoflagellates produce mainly low-polar analogs including CTX4A, CTX4B, CTX3C, and 49-*epi*CTX3C [5,10,11,12,13]. These compounds are metabolically oxidized to form analogs such as CTX1B, 54-deoxyCTX1B, 2,3-dihydroxyCTX3C, and 51-hydroxyCTX3C within the food web [1,14]. More than 20 analogs have been reported from the Pacific (Figure 1) [15].

The minimum amount of CTX required to induce intoxication has been estimated to be 10 mouse units (M.U.), which is equivalent to 70 ng of CTX1B [1]. The Food and Drug Administration of the United States (U.S. FDA) has established a guidance level for Pacific CTXs at 0.01 μg/kg CTX1B equivalent [16]. The European Food Safety Authority (EFSA) referred a concentration of 0.01 μg P-CTX-1 equivalents/kg fish, but they concluded that liquid chromatography-tandem mass spectrometry methods can be of value for the quantification of CTX-group toxins, but certified reference standards and reference materials need to be provided to allow for method development and (inter-laboratory) validation [17].

The traditional method for evaluating fish toxicities has been the mouse bioassay (MBA) [5,18,19,20]. However, due to the lack of specificity of MBA and concerns for animal welfare [6,21], alternative methods have become necessary. Several alternative methods to MBA have been developed including functional and immunological assays as well as chemical analysis [6]. Functional assays include the cell-based assay that uses mice neuroblastoma cell line, Neuro2a (N2a assay) [21,22], and the receptor binding assay that employs rat or porcine synaptosomes as receptors. This involves competition with tritium or fluorescent-labeled brevetoxin, which shares receptors with CTXs [23,24]. Among the immunological methods, sandwich ELISA (enzyme-linked immunosorbent assay) has been produced using specific antibodies against the left or right wings of CTX1B and CTX3C [25,26].

Regarding chemical methods, various research groups have reported the use of liquid chromatography-mass spectrometry (LC/MS) and liquid chromatography-tandem mass spectrometry (LC-MS/MS). The initial challenges were overcome by Lewis and coworkers using LC-MS [27] and LC-MS/MS [28]. In the LC-MS/MS analysis, they monitored ion transitions including [M+NH_4_]^+^ > [M-H_2_O]^+^, [M+NH_4_]^+^ > [M-2H_2_O]^+^, and [M+NH_4_]^+^ > [M-3H_2_O]^+^ of CTX1B (P-CTX) using a Sciex API-III system; however, the limit of detection (0.04 μg/kg) was not sufficient to meet the FDA guidance level (0.01 μg/kg) [28]. Wu et al. conducted a quantitative analysis of CTX1B (P-CTX-1), 52-*epi*-54-deoxyCTX1B (P-CTX-2), and 54-deoxyCTX1B (P-CTX-3) and monitored the respective transitions of [M+NH_4_]^+^ > [M-2H_2_O]^+^ using a Sciex 5500 QTRAP system with a turbo-ion spray. This resulted in an LOQ at 2.5 μg/kg of CTX1B in a flesh extract of a moray eel species, *Gymnothorax undulatus* [29]. Under electrospray ionization, CTXs provided [M+H]^+^, [M+NH_4_]^+^, and many of their dehydrated ions such as [M+H-H_2_O]^+^, [M+H-2H_2_O]^+^, [M+NH_4_-H_2_O]^+^, and [M+NH_4_-2H_2_O]^+^. In other words, the CTX molecules were spread into many ions, and it provided a low intensity for each ion [28].

Yogi et al. demonstrated that by using an Agilent 6460 Triple Quadrupole LC/MS system equipped with an Agilent Jet Stream electrospray ionization source and methanolic mobile phase, a dominant [M+Na]^+^ ion was formed for each analyzed CTX analog. Since [M+Na]^+^ was stable and no product ions were observed, the ion transition of [M+Na]^+^ > [M+Na]^+^ was monitored using high collision energy to achieve the highest S/N ratio. The LOD and LOQ for this method were determined to be 0.01 and 0.04 µg/kg, respectively [10,30]. This method was further optimized to improve the sensitivity for the detection of CTX analogs. Estevez et al. reported LOD (S/N > 3) and LOQ (S/N > 10) values of 0.0045 µg/kg and 0.0150 µg/kg, respectively [31].

In these reports, the reference materials used were CTX analogs purified from natural sources [29,32,33,34,35,36,37,38] or had been chemically synthesized [39]. Kato and Yasumoto prepared the reference materials of CTX analogs (JFRL-CTX-RM), quantified by the quantitative nuclear magnetic resonance (q-NMR) technique [40], and provided them to representative research institutions. Oshiro et al. prepared a solution containing CTX analogs (NIHS-CTX-Mix) and quantified them by JFRL-CTX-RM. They improved Yogi’s method using the same system and achieved the estimated LOD (S/N > 5) and LOQ (S/N > 10) for all CTX analogs of 0.001 µg/kg and 0.005 µg/kg, respectively [41,42]. However, this method of monitoring [M+Na]^+^ > [M+Na]^+^ transitions is only capable with equipment provided by certain manufacturers in low-level analysis.

Here, we present an improved CTX analysis method using an acetonitrile-based mobile phase supplemented with Na^+^ or Li^+^ that can potentially achieve the highly sensitive analysis of CTXs using LC-MS/MS equipment.

## 2. Results and Discussion

### 2.1. Preliminary Experiments with Previously Reported Methods

The LCMS-8060 triple quadrupole tandem mass spectrometer (Shimadzu Corporation, Kyoto, Japan) was used for preliminary studies to evaluate the system’s performance based on conditions reported in previous studies [10,29]. The CTX mix solution (JFRL-CTX-Mix) was prepared from five CTX reference materials: CTX1B, 52-*epi*-54-deoxyCTX1B, 51-hydroxyCTX3C, CTX3C, and CTX4A (JFRL-CTX-RMs), supplied by JFRL (Japan Food Research Laboratories, Tokyo, Japan). Full scan MS spectra for the five CTX analogs contained in the JFRL-CTX-Mix were acquired in the mass range *m*/*z* 900–1200. This was carried out using acetonitrile or methanol-based mobile phases as described in previous reports by Wu et al. [29] and Yogi et al. [10], respectively.

Under the acetonitrile-based mobile phase, [M+Na]^+^, [M+K]^+^, [M+NH_4_]^+^, [M+H-H_2_O]^+^, and [M+H-2H_2_O]^+^ ions were detected, as described in a previous paper (Appendix A) [29]. Under the methanolic condition, [M+Na]^+^ ions were the most abundant. However, peaks corresponding to [M+K]^+^, [M+NH_4_]^+^, [M+H-H_2_O]^+^, and [M+H-2H_2_O]^+^ were also detected (Appendix A). Contrary to previous results [10], it was not observed that the ions converged only on [M+Na]^+^, as reported by Tartaglione et al. [43]. To find suitable precursor and product ions for MRM analysis, several transitions were set for each analog. These were based on results obtained from the product ion scan analyses derived from the most abundant ions other than [M+Na]^+^ and the transitions reported in previous studies (Appendix A) [44]. In this experiment, six MRM transitions of each CTX were evaluated ([M+H]^+^ > fragment 1, [M+H]^+^ > fragment 2, [M+NH_4_]^+^ > fragment 1, [M+NH_4_]^+^ > fragment 2, [M+NH_4_]^+^ > [M+H]^+^, [M+NH_4_]^+^ > [M+H-H_2_O]^+^). Fragments 1 and 2 were *m*/*z* 125.1 and 155.1, respectively, for 52-*epi*-54-deoxyCTX1B, CTX3C, and CTX4A. Fragments 1 and 2 were *m*/*z* 121.0 and 149.1, respectively, for CTX1B and 51-hydroxyCTX3C. In the LC-MS/MS (MRM) chromatograms, the acetonitrile-based mobile phase (Appendix A) provided higher intensity and sharper peaks compared to those obtained under the methanolic mobile phase (Appendix A). Additionally, the acetonitrile condition provided superior separation, especially for low-polar analogs such as CTX3C and CTX4A (Appendix A).

However, the limit of detection (LOD) in both mobile phases was too high (e.g., the LOD of CTX1B was >0.5 ng/mL). Since these methods exhibit lower sensitivity compared to previous findings [10,29,41] and require further improvement, the following experiments were conducted aimed at ion-focusing generation using an acetonitrile-based mobile phase.

### 2.2. Production of [M+Na]^+^ or [M+Li]^+^ Ions Using Acetonitrile-Based Mobile Phase

We considered adding alkaline-metal ions to the mobile phase to favor the formation of the main adduct ions. As described above, an acetonitrile-based mobile phase was adopted because it showed better separation than a methanol-based mobile phase (Appendix A). Sodium hydroxide (NaOH) was used to generate [M+Na]^+^ ions because their solubility in organic solvents is high. Formic acid (0.1%) was added to provide an acidic mobile phase to protect the analytical column. Lithium hydroxide was also added since [M+Li]^+^ was considered to provide fragment ions suitable for MRM transitions such as in a previous report for palytoxin detection [45]. The concentration of metal ions in the mobile phase was set to be acidic, providing satisfactory results, as shown below. Thus, we continued the experiment without optimizing the metal ion concentrations.

To reduce the amount of CTX reference solution used, the optimization of MS conditions was carried out using SIM mode analysis instead of full SCAN mode analysis. The ions suspected of being generated such as [M+Li]^+^, [M+Na]^+^, [M+K]^+^, [M+H]^+^, [M+NH_4_]^+^, [M+H-H_2_O]^+^, and [M+H-2H_2_O]^+^ were monitored under mobile phase B including Na^+^ or Li^+^. When the mobile phase contained Na^+^ (NaOH), the detected ions converged to [M+Na]^+^. This led to an improvement in the intensities of [M+Na]^+^ by more than tenfold compared to the cases without NaOH (Figure 2a,b and Appendix A). Similarly, a mobile phase containing Li^+^ (LiOH) led to the generation of a high intensity of [M+Li]^+^ ions (Figure 2a,c and Appendix A).

To reduce the amount of JFRL-CTX-Mix or NIHS-CTX-Mix used, product ion scan analyses were carried out on a purified CTX3C reference sample obtained from natural sources. Product ion scan experiments were acquired using [M+Na]^+^ or [M+Li]^+^ precursor ions in collisional induced dissociation (CID) mode using collision energies of −20, −30, −40, −50, and −60 V (Figure 3). Although several product ions were observed from both precursor ions ([M+Na]^+^ or [M+Li]^+^) at higher CID voltages, the intensities of these ions were too low to employ for MRM transitions in low-level analysis. Consequently, we decided to monitor [M+Na]^+^ > [M+Na]^+^ or [M+Li]^+^ > [M+Li]^+^ in LC-MS/MS MRM mode to achieve a highly sensitive analysis. The CID energies and the other factors were optimized to provide the highest S/N ratios, as detailed in previous reports (Appendix A) [10,41,42].

The NIHS-CTX-Mix containing ca. 1 ng/mL of nine CTX analogs was used as the reference. The MS conditions LC-MS/MS in MRM-mode analysis were optimized. Monitoring each compound for the transition [M+Na]^+^ > [M+Na]^+^ using the mobile phase containing Na^+^, the intensities of six out of the nine CTX analogs (CTX1B, 52-*epi*-54-deoxyCTX1B, 54-deoxyCTX1B, 49-*epi*CTX3C, CTX4A, and CTX4B) were improved (LOD < 0.005 ng/mL, Figure 4A). However, the intensities of the remaining analogs, 2,3-dihydroxyCTX3C (LOD < 0.15 ng/mL), 51-hydroxyCTX3C, and CTX3C (LOD < 0.015 ng/mL), were extremely low (Figure 4A). In fish specimens from the Ryukyu Islands (Okinawa and Amami), Japan, only CTX1B analogs were present, which has been confirmed in previous reports [10,30,46,47,48,49]. Therefore, this method could be employed to assess the toxicity of specimens from regions such as the Ryukyu Islands where CTX1B analogs dominate.

In the mobile phase containing Li^+^, the MRM transition [M+Li]^+^ > [M+Li]^+^ was selected for monitoring. The intensities of the six CTX analogs above-mentioned were lower than those obtained with the mobile phase containing Na^+^ (Figure 4B). However, the intensities of all CTX analogs including the remaining three were similar (LOD < 0.005 ng/mL, Figure 4B). As a result, this method was deemed suitable for the simultaneous analysis of CTXs in specimens containing both CTX1B and CTX3C analogs.

The LC-MS/MS conditions were eventually established using an acetonitrile-based eluent supplemented with either Na^+^ or Li^+^, as shown in Appendix B.

### 2.3. Evaluation of a Quantitative Analysis Monitoring [M+Li]^+^ > [M+Li]^+^

A good chromatographic separation was achieved (Figure 5). The calibration curves for CTX1B ranged from 0.0022 to 0.110 ng/mL, exhibiting a good R^2^ value of over 0.999, and the S/N ratio of the lowest calibration points of each CTX analog was greater than 10 (Figure 6 and Figure 7). The LOQ determined from the lowest calibration point of the five analogs, CTX1B, 52-*epi*-54-deoxyCTX1B, 51-hydroxyCTX3C, CTX3C, and CTX4A, was found to be 0.0022 ng/mL, 0.0060 ng/mL, 0.0055 ng/mL, 0.0050 ng/mL, and 0.0055 ng/mL, respectively (Table 1). When 1 mL of the extraction solution was prepared from 5 g of fish tissue, the LOQ for the five analogs was 0.00040 μg/kg, 0.0012 μg/kg, 0.0011 μg/kg, 0.0010 μg/kg, and 0.0011 μg/kg, respectively. Thus, this method can be employed for performing FDA guidance level analysis (0.01 μg/kg CTX1B equivalent).

### 2.4. Analysis of Fish Flesh Extracts

From a methanol eluate prepared from the flesh of a *Gymnothorax javanicus* specimen, CTX1B (0.0005 μg/kg) with its two 54-deoxy analogs in unquantified levels and 51-hydroxyCTX3C (0.0016 μg/kg) were detected (Figure 8A, Table 2). Three CTX3C analogs were detected from the acetonitrile eluate. The detected analogs were 51-hydroxyCTX3C (0.0060 µg/kg), 49-*epi*CTX3C (0.0023 µg/kg), and CTX3C (0.0104 µg/kg) (Figure 8B, Table 2). The extract was previously analyzed using fluorescent ELISA for CTX3C (LOD = 0.00004 μg/kg; LOQ = 0.0001 μg/kg) and LC-MS/MS (LOD = 0.002 μg/kg; LOQ = 0.005 μg/kg), and CTXs were detected by ELISA at 0.003 μg/kg CTX3C equivalent, but nothing was detected using LC-MS/MS [50]. This result confirms that the method improved in this study will be useful for the analysis of CTXs at extremely low levels.

## 3. Conclusions

We optimized an improved method for analyzing CTXs using LC-MS/MS by monitoring the ion transitions of [M+Li]^+^ > [M+Li]^+^ in the mobile phases supplemented with trace amounts of lithium ions. The LOQs were determined from the lowest calibration curve concentration, created using q-NMR-quantified reference materials (JFRL-CTX-RM). The LOQ of the five CTX analogs ranged from 0.0020 to 0.0055 ng/mL or from 0.00044 to 0.0011 μg/kg when a 1 mL extract solution was prepared from 5 g of fish flesh. The performance of this method is suitable for analysis according to the U.S. FDA guidance level (0.01 µg/kg CTX1B equivalent). Further studies using mass spectrometers from other manufacturers will help establish it as a universal method for detecting extremely low levels of CTXs.

## 4. Materials and Methods

### 4.1. CTX Reference Materials

The q-NMR quantified reference materials (JFRL-CTX-RMs) of five ciguatoxin analogs were provided by JFRL [40]. These included CTX1B (43.3 ± 1.3 ng), 52-*epi*-54-deoxyCTX1B (58.4 ± 2.5 ng), 51-hydroxyCTX3C (45.3 ± 7.2 ng), CTX3C (38.5 ± 2.6 ng), and CTX4A (55.1 ± 5.2 ng). Each analog was dissolved in 1 mL of methanol (FUJIFILM Wako Pure Chemical Corporation, Osaka, Japan) to prepare the stock solution. The original mix solution (approximate concentration: 10 ng/mL, each) including five analogs was prepared from the five stock solutions in equal proportions, followed by dilution with methanol to the target concentrations (JFRL-CTX-MIX).

The CTX-Mix standard solution (NIHS-CTX-Mix ver.2) was prepared at the National Institute of Health Sciences (NIHS) using analogs purified or semi-purified from natural fish and dinoflagellate sources. These analogs were previously characterized by spectroscopic analysis in studies by Yasumoto and coworkers [15,33,34,35,51,52]. The mix solution comprised nine CTX analogs, and the concentration of each analog was quantified by JFRL-CTX-RM: CTX1B (4.0 ng/mL), 52-*epi*-54-deoxyCTX1B (5.8 ng/mL), 54-deoxyCTX1B (4.6 ng/mL), 51-hydroxyCTX3C (10.8 ng/mL), 2,3-dihydroxyCTX3C (11.2 ng/mL), 49-*epi*CTX3C (4.8 ng/mL), CTX3C (4.2 ng/mL), CTX4A (6.0 ng/mL), and CTX4B (4.4 ng/mL). Due to the lack of JFRL-CTX-RMs for 54-deoxyCTX1B, 2,3-dihydroxyCTX3C, 49-*epi*CTX3C, and CTX4B, these were quantified using 52-*epi*-54-deoxyCTX1B, 51-hydoxyCTX3C, CTX3C, and CTX4A, respectively.

### 4.2. Reagents

All organic solvents and formic acid were of LC-MS grade and were purchased from FUJIFILM Wako Pure Chemical Corporation (Osaka, Japan). Ammonium formate (Special Grade, ≥95.0%), lithium hydroxide monohydrate (for Amino Acid Automated Analysis), and 5 mol/L sodium hydroxide solution were purchased from FUJIFILM Wako Pure Chemical Corporation (Osaka, Japan). Ultra-pure water was prepared with a Milli-Q^®^ Integral Water Purification System (Millipore, Bedford, MA, USA).

### 4.3. LC-MS/MS Analysis

#### 4.3.1. Equipment

The analysis was performed by a liquid chromatograph coupled to a triple quadrupole tandem mass spectrometer, specifically an LCMS-8060 equipped with a Nexera™ X2 and an LCMS-8060NX equipped with a Nexera X3 UHPLC system (both from Shimadzu Corporation, Kyoto, Japan). The chromatographic separation was performed using Shim-pack™ Velox (50 mm × 2.1 mm I.D., 1.8 µm, Shimadzu Corporation, Kyoto, Japan) columns that use core shell technology bound with an octadecylsilyl base (ODS).

#### 4.3.2. Analytical Conditions of Preliminary Experiments

The LCMS-8060 triple quadrupole tandem mass spectrometer was used in the preliminary studies to assess performance based on the conditions reported in previous studies [10,29]. Using JFRL-CTX-MIX, the mass spectra of five CTX analogs were acquired in full scan mode using the acetonitrile or methanol-based mobile phases as described in those studies.

In the acetonitrile-based mobile phase, mobile phase A consisted of a 2 mmol/L ammonium formate aqueous solution, while phase B was acetonitrile [29]. The gradient elution program began at 50% B, moving up to 60% B in 1 min, followed by a two-phase linear gradient to 70% B in 6 min and to 90% B in 4 min while maintaining a flow rate of 0.4 mL/min. After an isocratic hold time of 1 min at 90% B, the gradient was returned to the initial condition of 50% B at 12.01 min. The total running time was 16 min. For the methanol-based condition, mobile phase A was a 5 mmol/L ammonium formate aqueous solution containing 0.1% formic acid, while mobile phase B was methanol [10]. The gradient elution program began at 78% B, followed by a linear gradient up to 88% B over 10 min at a flow rate of 0.4 mL/min. After an isocratic hold time of 4 min at 88% B, the gradient was returned to the initial condition of 78% B at 14.01 min. The total running time was 18 min. The following conditions were employed for both the acetonitrile and methanol-based mobile phases. The column oven was maintained at 40 °C, and the sample tube in an autosampler rack was cooled to 4 °C. A 1 µL portion of JFRL-CTX-MIX was injected. Mass spectra were acquired in full scan mode (Scan range *m*/*z* 900–1200 at a scan speed 769 u/s). The parameters were set as follows. Interface temperature: 300 °C, interface voltage: +4 kV, DL temperature: 250 °C, heat block temperature: 400 °C, nebulizer gas flow: 2 L/min, heating and drying gas flow: 10 L/min, CID gas pressure: 270 kPa.

Six MRM transitions were selected including those reported in earlier papers [39,44] as well as abundant ions in the product ion analysis that were derived from precursor ions other than [M+Na]^+^ (e.g., [M+H]^+^, [M+NH_4_]^+^) obtained in the scan measurements. MS parameters (various voltages, gas flow rates and gas temperatures) of the MRM measurement were optimized to obtain the highest intensity for each analog in the MRM chromatograms. Q1 pre-bias, collision energy, and Q3 pre-bias voltages were optimized by standard software (LabSolutions LCMS). In the acetonitrile-based condition, the same LC conditions were used for the spectral experiment described above (Appendix A) [29]. In the methanol-based condition, mobile phase A was a 5 mmol/L ammonium formate aqueous solution containing 0.1% formic acid, while mobile phase B was methanol. The gradient elution program began at 60% B, followed by a linear gradient up to 88% B in 10 min at the flow rate of 0.4 mL/min. After an isocratic hold time of 4 min at 88% B, the gradient was returned to the initial conditions of 60% B at 14.01 min. The total running time was 18 min (Appendix A) [10]. The following conditions were employed for both the acetonitrile and methanol-based mobile phases. The column oven was maintained at 40 °C, and the sample tube in an autosampler rack was cooled to 4 °C. A 5 µL portion of JFRL-CTX-MIX was injected. The ESI ion source and interface parameters were set for sensitive analysis optimized for ammonia adducts such as those associated with CTX1B. The parameters were set as follows. Interface temperature: 300 °C, interface voltage: +4 kV, DL temperature: 190 °C, heat block temperature: 400 °C, nebulizer gas flow: 3 L/min, heating and drying gas flow: 10 L/min, CID gas pressure: 270 kPa.

#### 4.3.3. Investigation into [M+Na]^+^ and [M+Li]^+^ Ions

Our system was unable to mainly detect sodium adduct ions using only methanol as the mobile phase. To achieve high sensitivity detection, Na^+^ or Li^+^ ions were added to the mobile phase with the aim of converging the CTX precursor ions into one. Alkali metal was introduced into the acetonitrile-based mobile phase since it exhibited better separation than the methanol-based mobile phase observed in the preliminary experiment (Section 4.3.2).

For the generation of [M+Na]^+^ ions, the mobile phases consisted of A: ultrapure water/formic acid (1000:1, *v*:*v*) and B: acetonitrile/formic acid/0.05 M sodium hydroxide aqueous solution (1000:1:1, *v*:*v*:*v*). For the generation of [M+Li]^+^ ions, the mobile phases consisted of A: water/formic acid (1000:1, *v*:*v*) and B: acetonitrile/formic acid/0.1 M lithium hydroxide monohydrate solution (1000:1:1, *v*:*v*:*v*). A similar gradient elution program was employed for both phases. The gradient began at 40% B with an isocratic hold time of 2.5 min at 40% B, followed by a linear gradient up to 85% B in 9.5 min at a flow rate of 0.4 mL/min. After an isocratic hold time of 5 min at 100% B from 12.01 to 17.00 min at a 0.6 mL/min flow rate, the gradient was returned to the initial condition of 40% B at 17.01 min. The total running time was 20 min. The column oven was maintained at 40 °C, and the sample tube in an autosampler rack was cooled to 4 °C. The injection volume was 5 µL of each sample solution. In the time program, only 2.5–12.5 min was introduced to MS. The triple quadrupole tandem mass spectrometer (LCMS-8060NX), equipped with a heated ESI source and ion focus capabilities, was used to detect [M+Li]^+^ > [M+Li]^+^ or [M+Na]^+^ > [M+Na]^+^ MRM transitions for nine compounds. The parameters were set as follows. Interface temperature: 350 °C, interface voltage: +1 kV, DL temperature: 300 °C, heat block temperature: 400 °C, nebulizer gas flow: 3 L/min, heating and drying gas flow: 15 L/min and 5 L/min, CE: -30 or 40 V, CID gas pressure: 270 kPa, ion focus voltage: +4 kV. The final LC-MS/MS analysis conditions for quantification are shown in Appendix B.

#### 4.3.4. Evaluation of Established LC-MS/MS Method for Monitoring [M+Li]^+^

The performance of the method for monitoring [M+Li]^+^ > [M+Li]^+^ was evaluated using JFRL-CTX-Mix. The JFRL-CTX-Mix was diluted with methanol to create a standard solution for the calibration curves at various levels: 0.1, 0. 02, 0.01, 0.005, and 0.002 ng/mL. The calibration curve range was determined to ensure that the repeatability of the peak area (*n* = 3) was less than 20%, and the average accuracy of each calibration point was 100 ± 5%. The limit of quantification (LOQ) for the five CTX analogs was defined as the minimum level of the calibration points. The S/N ratio was calculated using the RMS method utilizing noise from a 0.5 min interval near the peak.

#### 4.3.5. Analysis of Fish Flesh Extract

The fish used was a moray eel, specifically the *Gymnothorax javanicus* species, collected from Viti Levu Island in Fiji [41]. The flesh extract solution containing CTXs used in this study was the residual of the solution analyzed in a previous study and stored at −30 °C until used [41,50]. The methanol eluate may have contained analogs that include a diol moiety such as CTX1B, 52-*epi*-54-deoxyCTX1B, 54-deoxyCTX1B, and 1,2-dihydroxyCTX3C. Other analogs such as CTX4A, CTX4B, 49-*epi*CTX3C, and CTX3C may have been contained in the acetonitrile eluate [41,53].

Detected analogs were quantified using the calibration curves of five analogs contained in the JFRL-CTX-MIX. Due to the lack of RM, the quantification of 2,3-dihydroxyCTX3C, 54-deoxyCTX1B, and 49-*epi*CTX3C were carried out using the 51-hydroxyCTX3C, 52-*epi*-54-deoxyCTX1B, and CTX3C calibration curves, respectively.

## Figures and Tables

**Figure 1 toxins-16-00170-f001:**
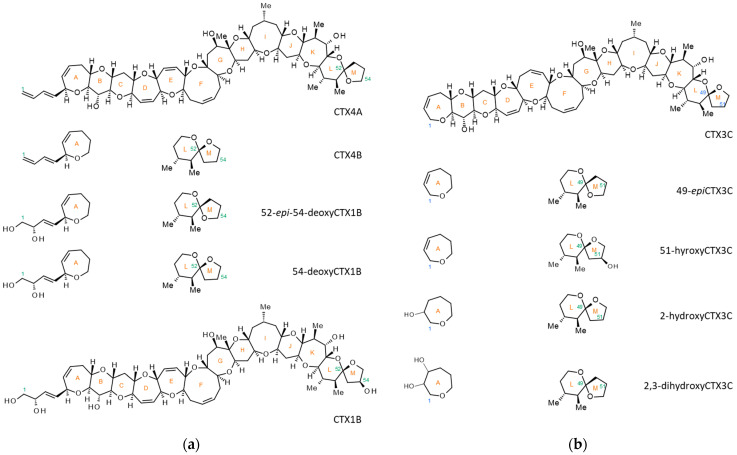
The structures of representative CTXs from the Pacific. (**a**) CTX1B analogs and (**b**) CTX3C analogs.

**Figure 2 toxins-16-00170-f002:**
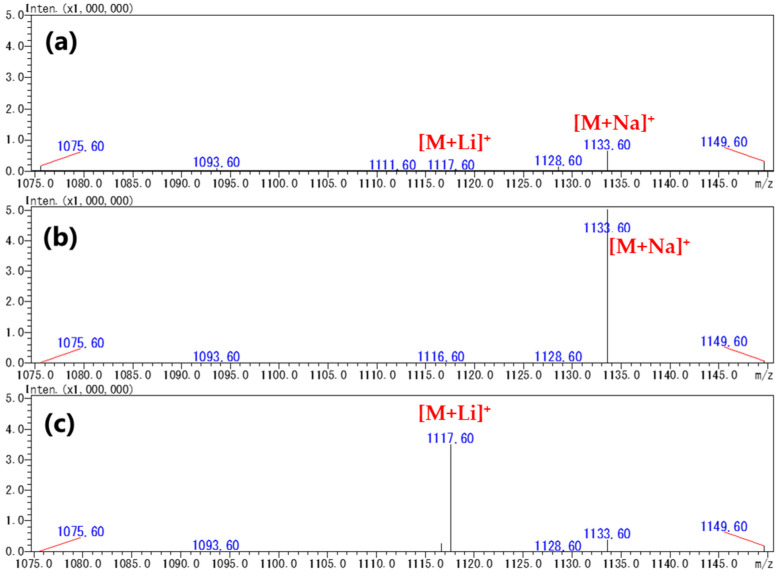
Mass spectra of CTX1B acquired under acetonitrile-based mobile phases (**a**) without any supplement, (**b**) supplemented with Na^+^ (NaOH), and (**c**) supplemented with Li^+^ (LiOH). The spectra of the other analogs are shown in Appendix A.

**Figure 3 toxins-16-00170-f003:**
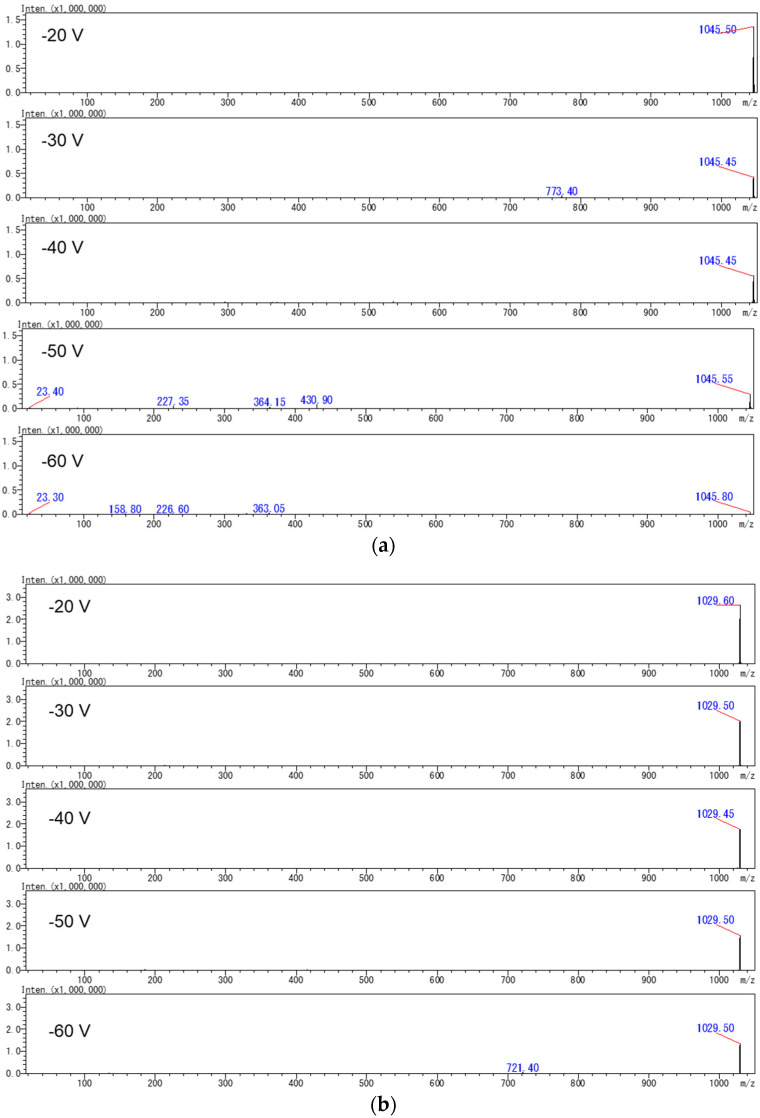
Product ion spectra of CTX3C derived from (**a**) *m*/*z* 1045.50 ([M+Na]^+^) and (**b**) *m*/*z* 1029.50 ([M+Li]^+^) under different CID energies. The mobile phases were supplemented with (**a**) Na^+^ and (**b**) Li^+^.

**Figure 4 toxins-16-00170-f004:**
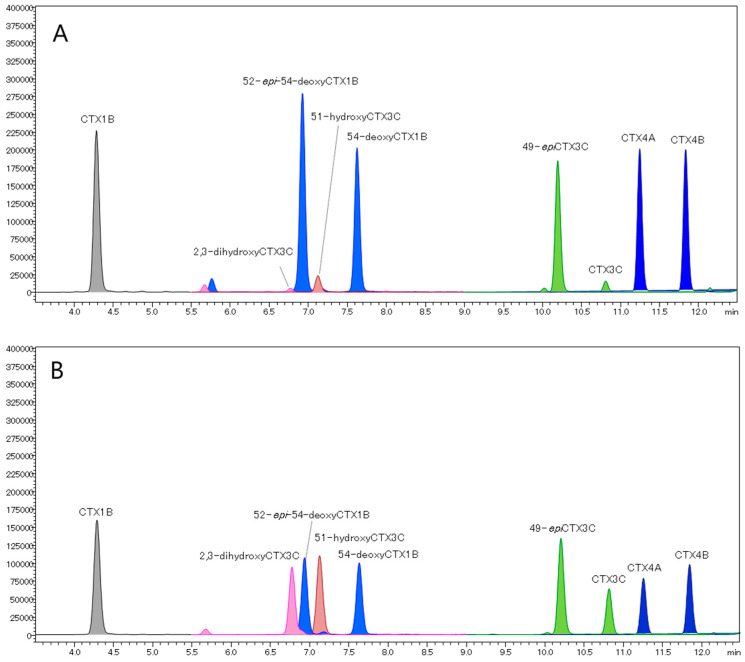
LC-MS/MS (MRM mode) chromatograms of the NIHS-CTX-Mix comprised of CTX1B (1.00 ng/mL), 2.3-dihydroxyCTX3C (2.80 ng/mL), 52-*epi*-54-deoxyCTX1B (1.45 ng/mL), 51-hydroxy-CTX3C (2.70 ng/mL), 54-deoxyCTX1B (1.15 ng/mL), 49-*epi*CTX3C (1.20 ng/mL), CTX3C (1.05 ng/mL), CTX4A (1.50 ng/mL), and CTX4B (1.10 ng/mL). Analyses were carried out with acetonitrile-based mobile phases containing (**A**) Na^+^ and (**B**) Li^+^ and MRM transitions were set as [M+Na]^+^ > [M+Na]^+^ and [M+Li]^+^ > [M+Li]^+^, respectively. The LC-MS/MS conditions are shown in Appendix B. (**A**) [M+Na]^+^ > [M+Na]^+^: CTX1B *m*/*z* 1133.60 > 1133.60, 2.3-dihydroxyCTX3C *m*/*z* 1079.60 > 1079.60, 52-*epi*-54-deoxyCTX1B and 54-deoxyCTX1B *m*/*z* 1117.60 > 1117.60, 51-hydroxyCTX3C *m*/*z* 1061.60 > 1061.60, 49-*epi*CTX-3C and CTX-3C *m*/*z* 1045.60 > 1045.60, CTX4A and CTX4B *m*/*z* 1083.60 > 1083.60. (**B**) [M+Li]^+^ > [M+Li]^+^: CTX1B *m*/*z* 1117.60 > 1117.60, 2.3-dihydroxyCTX3C *m*/*z* 1063.60 > 1063.60, 52-*epi*-54-deoxyCTX1B and 54-deoxyCTX1B *m*/*z* 1101.60 > 1101.60, 51-hydroxyCTX3C *m*/*z* 1045.60 > 1045.60, 49-*epi*CTX3C, and CTX3C *m*/*z* 1029.60 > 1029.60, CTX4A and CTX4B *m*/*z* 1067.60 > 1067.60.

**Figure 5 toxins-16-00170-f005:**
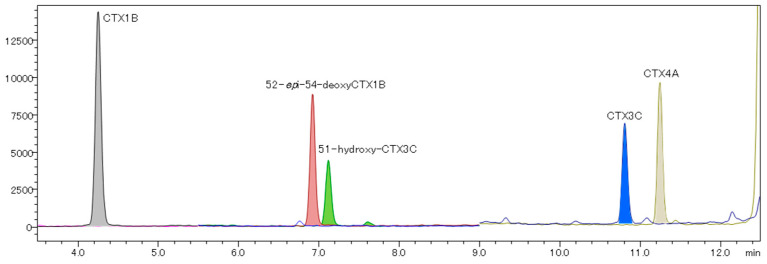
The LC-MS/MS (MRM mode) chromatogram monitoring [M+Li]^+^ > [M+Li]^+^ of the JFRL-CTX-Mix (0.1 ng/mL).

**Figure 6 toxins-16-00170-f006:**
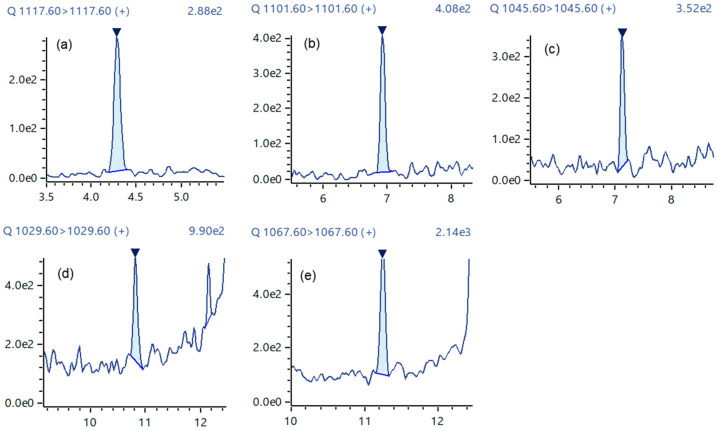
LC-MS/MS chromatograms at the lowest calibration point of CTX analogs in the JFRL-CTX-Mix. (**a**) CTX1B: 0.0022 ng/mL, (**b**) 52-*epi*-54-deoxyCTX1B: 0.0060 ng/mL, (**c**) 51-hydroxyCTX3C: 0.0055 ng/mL, (**d**) CTX3C: 0.0050 ng/mL, and (**e**) CTX4A: 0.0055 ng/mL.

**Figure 7 toxins-16-00170-f007:**
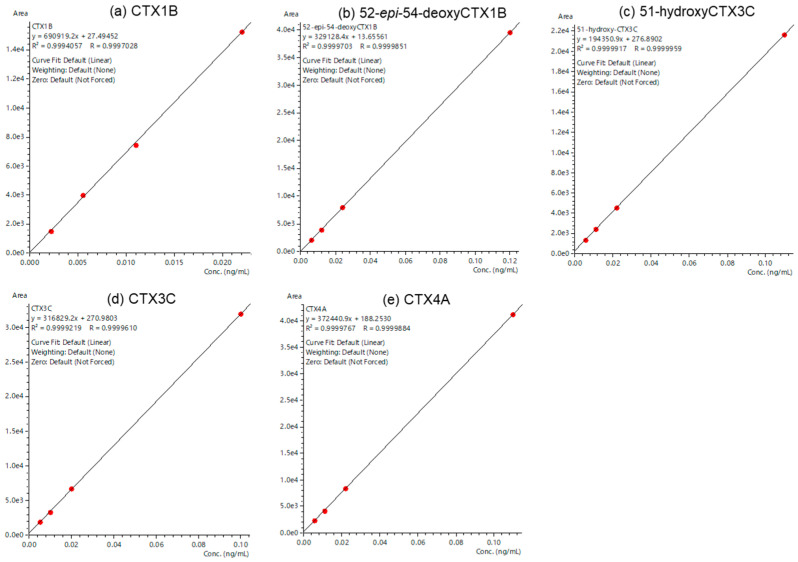
The calibration curves of the CTX analogs. (**a**) CTX1B (0.0022–0.110 ng/mL), (**b**) 52-*epi*-54-deoxyCTX1B (0.0060–0.120 ng/mL), (**c**) 51-hydroxyCTX3C (0.0055–0.110 ng/mL), (**d**) CTX3C (0.0050–0.100 ng/mL), (**e**) CTX4A (0.0055–0.110 ng/mL).

**Figure 8 toxins-16-00170-f008:**
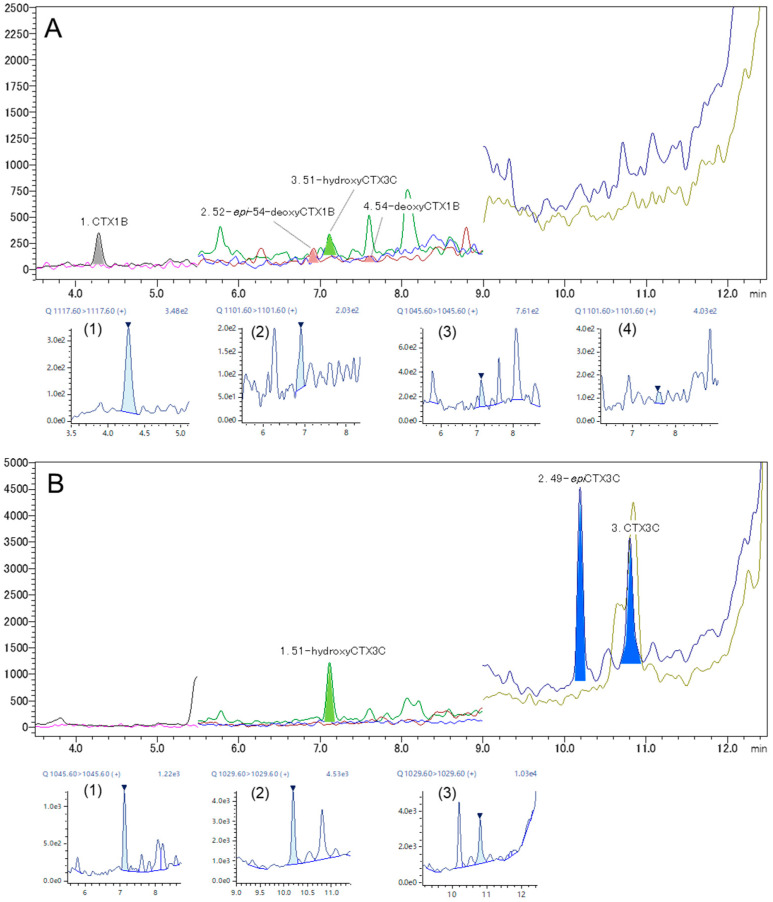
LC–MS/MS chromatograms of the extract prepared from a flesh sample of *Gymnothorax javanicus*. (**A**) methanol eluate and (**B**) acetonitrile eluate.

**Table 1 toxins-16-00170-t001:** Evaluation results for the lowest point on the calibration curve (LOQ).

CTX Analog	LOQ (ng/mL)	LOQ ^1^ (μg/kg)	Retention Time (min)	%RSD ^2^ (Area)	Accuracy ^2^ (%)	S/N ^2^
CTX1B	0.0022	0.00044	4.289	11.02	98	81
52-*epi*-54-deoxyCTX1B	0.0060	0.0012	6.931	5.38	105	64
51-hydroxy-CTX3C	0.0055	0.0011	7.121	8.76	103	22
CTX3C	0.0050	0.0010	10.815	12.32	102	22
CTX4A	0.0055	0.0011	11.250	17.28	105	36

^1^ LOQ when the solution was prepared at 5 g flesh equivalent/mL. ^2^ Results obtained at LOQ levels. Column: Shim-pack Velox C18 (50 mm × 2.1 mm I.D., 1.8 μm). Mobile phase A: water/formic acid (1000:1, *v*:*v*). Mobile phase B: acetonitrile/formic acid/0.1 M lithium hydroxide monohydrate solution (1000:1:1, *v*:*v*:*v*).

**Table 2 toxins-16-00170-t002:** The contents of CTX analogs (µg/kg) in the fish extracts from the flesh of *Gymnothorax javanicus* (methanol eluate and acetonitrile eluate).

Eluate ^1^	CTX1B	2,3-diOH-CTX3C ^2^	*epi*-Deoxy-CTX1B ^3^	Deoxy-CTX1B ^4^	51-OH-CTX3C ^5^	49-*epi*CTX3C	CTX3C
MeOH ^6^	0.0005	-	<LOQ ^8^	<LOQ ^8^	0.0016	-	-
ACN ^7^	-	-	-	-	0.0060	0.0023	0.0104

^1^: Eluate from the primary-secondary amine (PSA) cartridge. Analogs containing the diol moiety might be contained in the methanol eluate, and the others were in acetonitrile eluate. ^2^: 2,3-dihydroxyCTX3C. ^3^: 52-*epi*-54-deoxyCTX1B. ^4^: 54-deoxyCTX1B. ^5^: 51-hydroxyCTX3C. ^6^: methanol. ^7^: acetonitrile. ^8^: detected, but the level was lower than the LOQ.

## Data Availability

The data presented in this study are available on request from the corresponding authors.

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
