# Peer review of "Detection of Extremely Low Level Ciguatoxins through Monitoring of Lithium Adduct Ions by Liquid Chromatography-Triple Quadrupole Tandem Mass Spectrometry"

_toxins, 2024, doi:10.3390/toxins16040170_

Round 1

Reviewer 1 Report

Comments and Suggestions for Authors

Was only one eel tested for this manuscript...I would have preferred to see at least 10 and some other fish species to determine if this method works on other types of tissue.

Author Response

Dear Reviewer 1

The authors would like to appreciate the reviewers for their valuable comments and suggestions to improve the quality of the manuscript.

Please confirm the word file for detailed corrections.

Have a nice new year.

Reviewer 2 Report

Comments and Suggestions for Authors

This paper describes a LC-MS/MS paper for the detection of CTXs using adduct mass spectrometry. Whilst interesting there are a number of CTX LC-MS/MS papers published. The main issue is that, as far as I am aware, the P-CTX group fragment nicely under MS/MS conditions to produce characteristic daughter ions from the M+H+ parent ion of m/z 125 and 155 as described in numerous papers (e.g. see Murray et al). In our experience these transitions are highly sensitive and specific. These conditions have not been examined, or even discussed in this paper and should be included for comparison. The instrument that the authors use should be capable of these experiments and shoudl be tested, and reported even if the results are negative.

Author Response

Dear Reviewer 2

The authors would like to appreciate the reviewers for their valuable comments and suggestions to improve the quality of the manuscript.

Please confirm the word file for detailed corrections.

Have a nice new year.

Reviewer 3 Report

Comments and Suggestions for Authors

The manuscript entitled “Detection of Extremely Low Level Ciguatoxins through Monitoring of Lithium Adduct Ions by Liquid Chromatography-Triple Quadrupole Tandem Mass Spectrometry”

aims to propose an analytical strategy to increase sensitivity in LC-MS/MS detection of Ciguatoxins (CTXs). The Authors added  sodium hydroxide or lithium hydroxide to mobile phases (not clear if a proper optimization testing different concentrations of cations has been conducted or not) trying to obtain intense [M+Na]+ or [M+Li]+ adduct ions in the perspective to select MRM transitions to be used to detect and quantify trace levels of CTXs.

Although Ciguatera Poisoning is a global issue and so, in principle, the objective of the study is quite interesting, I have the feeling that the final proposed conditions have been selected not conducting a proper optimization. Some recent papers describing the use of chromatographic conditions have been not cites and so not taken into consideration. Furthermore, I consider the discussion incomplete, with the matrix effect not discussed at all and a lot of very generalist statements. The manuscript appears more like a draft to be further refined than a final version to be submitted.

 I strongly suggest the Authors to improve the quality of the manuscript.

I have attached the pdf file of the manuscript containing lots of specific comments.

Comments on the Quality of English Language

I'm not a native speaker but It's clear to me that English requires some refinements

Author Response

Dear Reviewer 3

The authors would like to appreciate the reviewers for their valuable comments and suggestions to improve the quality of the manuscript.

Please confirm the word file for detailed corrections.

Have a nice new year.

Round 2

Reviewer 1 Report

Comments and Suggestions for Authors

My only issue is that more low level fish were not sampled using this method.  Was this method tested on gambierdiscus cultures for detection of these analogs as well?

Comments on the Quality of English Language

there are some minor grammatical errors that should be cleaned up, but the manuscript reads and flows well

Author Response

The authors would like to appreciate the reviewer for your valuable comments and suggestions to improve the quality of the manuscript. 

We are confident that this method can be used for CTX analysis with general LC-MS/MS from any manufacturer.
It is expected that this method can be used to analyze the samples you mentioned.

Basically, this manuscript has undergone a native check and we have obtained an English proofreading certificate.

We decided that we don't need to revise the manuscript, so we will resubmit it again.

Thank you very much for your review.

Reviewer 3 Report

Comments and Suggestions for Authors

Page 1, line 37: The specific skeletal structures of CTXs depend on the sea area.

I’m suggesting again to delete this sentence because the use of the regional naming convention for these compounds (pacific CTXs, Indian CTXs, and so on) makes sense only to maintain consistency in the literature. Still, these toxins are unlikely to be restricted to the Caribbean, Pacific or Indian.

Page 1, line 41: Turn the sentence into: “The structure of this last CTXs series, however…..”

Page 2, line 64: Although I already suggested to change the sentence: “However, in addition to its low sensitivity, specificity, and concerns for animal welfare [6][21], alternative methods have become necessary.”, the Authors didn’t agree. I strongly suggest changing the sentence into:

“However, due to not specificity of MBA and concerns for animal welfare [6][21], alternative methods have become necessary.”

Page 3, line 89: Please turn the sentence “Yogi et al. discovered that ions were focused as [M+Na]+ under the methanolic mobile phase, which was analyzed using an Agilent 6460 Triple Quadrupole LC/MS system equipped with an Agilent Jet Stream electrospray ionization source.” Into:

“Yogi et al. (year) demonstrated that using an Agilent 6460 Triple Quadrupole LC/MS system equipped with an Agilent Jet Stream electrospray ionization source and  methanolic mobile phase  a dominant [M+Na]+ ion is formed for each analyzed CTX.”

Page 4, line 126: The sentence is not complete, if Authors refer to Tartaglione et al. they should say Tartaglione et al (2023) and also include the complete reference Tartaglione et al. 2023, in the References list: (https://www.sciencedirect.com/science/article/pii/S0045653523002072)

Page 4, line 147: Please turn the sentence “We considered adding alkalin-metal ions to the mobile phase to faver the formations of a main adduct ions.

 Into: “We considered adding alkali-metal ions to the mobile phase to favor the formations of main adduct ions.”

Page 4, 147-162 : Although already asked for further details about How the Authors have selected the right concentration of NaOH (and LiOH) to add to the mobile phase? No experiments were conducted to obtain the optimal conditions for the cationization. The Authors didn’t give an explanation.

Page 7, Figure 4: Please turn m/z into italics everywhere.

Author Response

Page 1, line 37: The specific skeletal structures of CTXs depend on the sea area.

I’m suggesting again to delete this sentence because the use of the regional naming convention for these compounds (pacific CTXs, Indian CTXs, and so on) makes sense only to maintain consistency in the literature. Still, these toxins are unlikely to be restricted to the Caribbean, Pacific or Indian.

Following the reviewer’s suggestion, the sentence was revised as “Based on skeletal structures of CTX analogs, they are classified into four groups, including CTX4A, CTX3C, C-CTX (Caribbean CTX), and I-CTX (Indian Ocean CTX) derivatives [4-6].” (L37-39)

Page 1, line 41: Turn the sentence into: “The structure of this last CTXs series, however…..”

To make clear that the I-CTX analogue was mentioned, the sentence was revised as “Of which, the structure of I-CTX has not yet been elucidated” (L39-L40).

Page 2, line 64: Although I already suggested to change the sentence: “However, in addition to its low sensitivity, specificity, and concerns for animal welfare [6][21], alternative methods have become necessary.”, the Authors didn’t agree. I strongly suggest changing the sentence into:

“However, due to not specificity of MBA and concerns for animal welfare [6][21], alternative methods have become necessary.”

Following the reviewer’s suggestion, the sentence was revised as “However, due to not specificity of MBA and concerns for animal welfare [6][21], alternative methods have become necessary.” (L63-64)

Page 3, line 89: Please turn the sentence “Yogi et al. discovered that ions were focused as [M+Na]+ under the methanolic mobile phase, which was analyzed using an Agilent 6460 Triple Quadrupole LC/MS system equipped with an Agilent Jet Stream electrospray ionization source.” Into:

“Yogi et al. (year) demonstrated that using an Agilent 6460 Triple Quadrupole LC/MS system equipped with an Agilent Jet Stream electrospray ionization source and methanolic mobile phase a dominant [M+Na]+ ion is formed for each analyzed CTX.”

 Following the reviewer’s suggestion, the sentence was revised as “Yogi et al. demonstrated that using an Agilent 6460 Triple Quadrupole LC/MS system equipped with an Agilent Jet Stream electrospray ionization source and methanolic mobile phase, a dominant [M+Na]+ ion was formed for each analyzed CTX analog.“ The published year was omitted as following the manuscript published in Marine Drugs. (L87-L89)

Page 4, line 126: The sentence is not complete, if Authors refer to Tartaglione et al. they should say Tartaglione et al (2023) and also include the complete reference Tartaglione et al. 2023, in the References list: (https://www.sciencedirect.com/science/article/pii/S0045653523002072)

The Tartaglione et al (2023) was cited and listed as reference #43 in the revised manuscript. (L125)

Page 4, line 147: Please turn the sentence “We considered adding alkalin-metal ions to the mobile phase to faver the formations of a main adduct ions. Into: “We considered adding alkali-metal ions to the mobile phase to favor the formations of main adduct ions.”

 Following the reviewer’s suggestion, the sentence was revised as “We considered adding alkali-metal ions to the mobile phase to favor the formations of main adduct ions.” (L145-L146)

Page 4, 147-162 : Although already asked for further details about How the Authors have selected the right concentration of NaOH (and LiOH) to add to the mobile phase? No experiments were conducted to obtain the optimal conditions for the cationization. The Authors didn’t give an explanation.

As a reviewer mentioned, we did not optimize the best concentration of the metal ions. Firstly, we adopted 0.1% formic acid with mobile phase as our common way. We just tried to supplement metal ion at the concentration that the mobile phase to be acidic. At that condition, we got results and we do not optimize the concentration of the metal ions. The purpose of this manuscript is to demonstrate that supplementing Li ion with mobile phase made it possible to perform an analysis to detected extremely low-level using equipment other than Agilent's.

We revised the sentence as “The concentration of metal ions in the mobile phase was set to be acidic, providing satisfactory results as shown below. Thus, we continued the experiment without optimizing the metal ion concentrations.” (L152-154)

Page 7, Figure 4: Please turn m/z into italics everywhere.

All m/z was turned into italics. In addition, hyphens were removed from acronyms of CTXs (e.g., CTX-1B to be CTX1B)